# Finding a Secure Place in the Home during the First COVID-19 Lockdown: A Pattern-Oriented Analysis

**DOI:** 10.3390/bs13010009

**Published:** 2022-12-22

**Authors:** Tamás Martos, Viola Sallay, Silvia Donato

**Affiliations:** 1Institute of Psychology, University of Szeged, 6720 Szeged, Hungary; 2Department of Psychology, Università Cattolica del Sacro Cuore–Milano, 20123 Milan, Italy

**Keywords:** COVID-19 pandemic, home, environmental self-regulation, personal niches, Emotional Map of the Home Interview, well-being

## Abstract

In challenging times, home is frequently the primary basis of environmental self-regulation processes, individual and relational coping, and well-being. This study aimed to identify multiple types of security experiences at home during the first lockdown period of the COVID-19 pandemic. We used data from 757 Hungarian adults who completed the online, modified form of the Emotional Map of the Home Interview method in 2020 after the outbreak of the COVID-19 pandemic. Participants imagined their homes, chose the place of security in their homes and rated their personal experiences (i.e., experiences of agency, communion, self-recovery, and distress) related to these places. Latent profile analysis of personal experiences revealed four types of relational-environmental self-regulation in secure places: “security in active self-recovery,” “security in detachment,” “security in doing and feeling good enough,” and “security in stress and compensation.” Profile membership was predicted by age, gender, and indices of psychological support and well-being. Results suggest that finding psychological security in the home is a multifaceted phenomenon that may be partly affected by the perception of the broader social-ecological context. Identifying subpopulations vulnerable to the challenges of the pandemic may help researchers and practitioners provide better support in times of local and global crises.

## 1. Introduction

The outbreak of the COVID-19 pandemic has exerted a considerable impact on various aspects of life, including mental health [1]. The experience of psycho-pathological symptoms, such as insecurity, anxiety, and hopelessness, significantly increased during the first wave of the pandemic [2]. Since the outbreak of the pandemic, several studies have explored the social-ecological aspects of coping with the uncertainty caused by the pandemic. More resilient relationships [3,4] and a social environment that supported basic psychological need satisfaction were associated with higher well-being [5,6]. Moreover, Liu and colleagues found gender differences in China, where post-traumatic stress symptoms (PTSS) were higher in women [7]. If we look at broader ecological aspects, negative mental health impact was more severe under poorer housing conditions [8] and, generally, in low and middle-income countries [9]. These results indicate that social-ecological factors, including the experiences in the home, counted significantly in the mental health outcomes of the pandemic [10]. In the present study, we aim to understand the various patterns of how people found emotional security in their homes during the first wave of the COVID-19 pandemic in Hungary.

### 1.1. Home as a Place of Security in the Pandemic

COVID-19-related challenges generated an unprecedented global level of insecurity for many people. We can assume that coping with these challenges might have been connected to altered home experiences. However, little research addressed the specific role of homes and home-related social-cognitive processes in coping with the pandemic. In a longitudinal study during the first wave of the pandemic, a higher level of initial attachment to the home predicted lower levels of depression, anxiety, and stress in two- and four-week follow-ups [11]. Moreover, homes with natural views from the window and pieces of the natural environment close to them were associated with better mental health, even for economically disadvantaged respondents in the pandemic [12,13]. Other research shows that the adverse effects of lockdown amplified the mostly hidden, dark side of home experiences and might have contributed to the loss of personal control and the decreased possibility of finding security and a safe place. Findings showed that lockdown fueled the increase in loneliness, especially among younger women; the level of loneliness was mainly unrelated to the amount of household work (i.e., activity level) [14]. Moreover, women experienced higher-than-average negative mental health issues, especially in vulnerable statuses, such as pregnancy and intimate partner violence [7,15]. In sum, in most cases, the home served as a place for restoration and well-being and could be perceived as a safe haven where the virus was least able to penetrate (compared to work or other outdoor spaces). However, the home could also be increasingly perceived as a “trap” for many during the pandemic [16,17].

### 1.2. General Framework: Home as a Personal Niche

The concept of personal niches and environmental self-regulation in the personal niche provide a framework in which we can interpret these different results. Moreover, they show potential paths to better understand unexplored aspects of home experiences in times of global challenges and insecurity. Throughout their lives, people strive to find self-congruent environments [18,19], also called personal niches [20,21], where they can thrive and accomplish meaningful pursuits [22,23]. In this regard, home is one of the vital personal niches in life and one of the main domains of self-accomplishment. Ideally, the home serves as a source of restoration. As such, it plays an essential part in one’s environmental self-regulation.

Environmental self-regulation is when people involve physical spaces and places to create their emotional responses and modify their ongoing emotions in response to rapidly changing environmental and social stimuli [24]. According to the hypothesis of environmental self-regulation, self-regulation and emotion-regulation processes involve not only mental, physical, and social strategies but also environmental strategies in which favorite places play an essential role [25]. In research about restorative environments, home environments and specific parts of the respondents’ homes are frequently mentioned as favorite places people cherish in their daily lives and visit when they need to replenish and relax [26].

The main aspects of environmental self-regulation include the potential for autonomous and competent activity [11] and good relationships (corresponding to the socio-physical nature of the home environment (c.f., [27])). Moreover, environmental self-regulation also involves the emotional experiences in the places which provide the necessary feedback for the person. Concerning favorite places, Korpela and colleagues [28] showed that visits to these places might affect emotional place experiences and mental well-being, partly by transforming negative cognitions and feelings into positive ones. Interestingly, respondents also reported distress experiences in their favorite place (e.g., loss of self-confidence). Although these experiences were much less common on average than positive experiences, they did show a significant negative relationship with life satisfaction and subjective health status. However, skillful environmental self-regulation can contribute to enduring well-being [28,29].

The potential variety of home experiences in the pandemic indicates multiple emotional meanings attached to the home. If we want to transform these general associations into the self-regulation processes in the home as a personal niche during the pandemic, we may assume that homes provide a secure base for individual and relational coping with challenges. At the same time, the self-regulation of stress might have been associated with temporary episodes of despair, doubts, loneliness, and insecurity. Beyond idealistic notions, real-life homes are emotionally patterned spaces where people have several positive and negative experiences that may connect to specific parts of the home at specific times [30]. In this regard, the home as a place of security and, more specifically, the place of security in the home may have a specific role in self-regulation outcomes.

### 1.3. The Present Study

From 11 March to 17 May 2020, the Hungarian government declared a state of emergency throughout the country due to the COVID-19 pandemic and ordered several measures to block the spread of the infection. The measures included restrictions on citizens leaving their homes, commonly called ‘lockdown.’ Previous research in Hungary [31,32,33] investigated the widespread impact of the COVID-19 pandemic and the resulting social changes on physical and mental health. However, these studies did not focus on specific self-regulation processes and the data were mainly published in Hungarian.

In the present research, we explore self-regulation strategies in the home during the first lockdown period of the pandemic in Hungary. More specifically, we investigate how emotional, behavioral, and relational experiences related to the place of security form meaningful configurations in the subjective worlds of the respondents. These configurations may represent specific environmental self-regulation strategies for searching for a physically and emotionally secure personal niche amidst the insecurity the COVID-19 pandemic and the resulting measures might have created. The specific aims of the present research are manifold.

First, we focus on security-related experiences since security is one of the fundamental human strivings and the one that was under challenge during the pandemic in a previously unprecedented manner [34]. We assumed that specific niche construction processes would characterize self-regulation in the place of security and, therefore, contribute to coping with pandemic-related stress and well-being.

Second, we aimed to assess four niche construction processes correspondingly. Two are organized around the person’s striving for agency and communion [35] that shape the place experience. The two other processes entail emotional states as feedback loops in the process of self-regulation [36]; they include self-recovery and distress, as described by Korpela and colleagues [28]. One specific interest concerned the occurrence of negative emotional experiences in the place of security. Emotional states such as sadness, low self-confidence, and stress were documented in previous research (c.f., [28]). Theoretically, favorite places’ role in self-regulation supports better mental health and restorative experiences. Nevertheless, self-regulation is a dynamic process [29], and negative aspects of emotional experiences in the favorite place may represent a specific type of person-environment congruence. Understanding the role of negative experiences may add to our understanding of environmental self-regulation.

Third, we adapt the Emotional Map of the Home Interview (EMHI) for our study, using a new procedure variation. Initially, the EMHI was a qualitative in-depth interview schedule designed to assess the patterned nature of home experiences in their spatial and temporal distribution. Sallay and colleagues [30] presented this unique procedure to explore place-related experiences, meaning-making processes, and relationship processes. We intended to introduce an online, quantified version of the procedure in the present study.

Fourth, we adapted a pattern-oriented approach to place experiences. Few studies have examined person–environment patterns in general. Moreover, no research has adopted a pattern-oriented framework (also known as a person-oriented or person-centered approach; [37]) as a theoretical and data analysis approach to home experiences. The pattern-oriented approach treats individual characteristics (variables) as interdependent parts of an integrated, holistic function [38]. In finding secure places at home, individuals’ place-related experiences form an interrelated, integrated field of experiences. The characteristics of these experiences can form specific configurations (types), i.e., nonlinear combinations of characteristics. Knowing these configurations can identify respondent groups with similar secure place experiences in their homes. Thus, the pattern-oriented approach offers a reasonable way to study complex sets of individual characteristics [39].

Finally, we want to explore the broader context of secure place experiences at home: their associations with basic social-physical characteristics of the home (e.g., floor area or the number of people living there) and the psychological conditions of good psychological functioning. For the latter, we used two sets of constructs. The satisfaction of basic psychological needs of autonomy, competence, and relatedness was previously found to be conducive to better functioning [5,6]. Moreover, higher life satisfaction was recently theorized to be a potential contextual predictor of increased environmental self-regulatory activity and its better emotional outcomes [28]. Accordingly, we assumed that secure place experiences were connected to the person’s cognitive evaluation of her resources in life and at home: life satisfaction [40] and satisfaction with home [41].

While person-oriented studies typically do not form explicit hypotheses due to the emergent nature of the patterns and the potential nonlinear character of their associations with external variables, we formed a few general assumptions. Specifically, we expected that a higher level of agentic and relational engagement (i.e., agency and communion), and affective experiences of restoration, along with low distress in the place of security in the home, would be predicted by a higher level of basic psychological need satisfaction, life satisfaction and satisfaction with the home during the pandemic.

## 2. Materials and Methods

### 2.1. Procedure

We conducted a cross-sectional online questionnaire study in Hungary with a convenience sample of voluntary Hungarian-speaking participants. The data were collected in May 2020 during Hungary’s first lockdown period of the COVID-19 pandemic. An online, anonymous questionnaire was designed on the Limesurvey platform and used for the survey. The questionnaire was distributed online on several social media sites (e.g., Facebook) and personal networks with the participation of psychology students at the University of Szeged, who received course credit for distributing the survey link. The research received ethical approval from the Joint Committee on Research Ethics in Psychology (EPKEB 2020-42).

In sum, 1,074 potential respondents opened the online survey, and 787 completed the questionnaire and submitted responses. Only adults (age > 18 years) without current psychiatric diagnoses were allowed to complete the study by the survey algorithm. Inclusion criteria were assessed by self-declaration. Eighteen respondents were ineligible since they did not respond about their secure place experiences. Therefore, we used a sample of 769 respondents for further analysis.

The respondents were informed that the research aimed to study “people’s everyday experiences during the lockdown in their homes.” They were assured that they could complete the questionnaire anonymously and that the data would be treated confidentially. The participants gave informed consent by accepting the introductory information and filling in the questionnaire. The respondents received no monetary compensation for their contribution.

### 2.2. Participants

The study sample is drawn from the online convenience community sample with 769 cases. In total, 12 cases were excluded: 8 as a multivariate outliers based on the Mahalanobis distance of the psychological variables of the place of security with a p-value of 0.01, and 4 cases after inspection of the short text answers to open-ended questions due to nonsense responses. The final sample for the analyses comprised 757 Hungarian respondents (mean age = 38.03 ± 12.31 years, ranging from 18 to 76 years; in 10 cases, age was missing). Concerning the appropriate sample size for an exploratory person-oriented (i.e., latent profile analysis) study, a sample size with N > 500 can be considered sufficient to accurately identify the correct number of latent profiles (c.f., [42]).

Table 1 presents the detailed sociodemographic characteristics of the sample. More information on details of housing (floor area, people currently living in the home) can be found in Table 2.

### 2.3. Measures

In the present study, we aimed to explore the patterns of finding security at home, the secure-place-related processes (agency and communion), and emotional experiences. Moreover, we assumed that the emergent patterns would connect with the broader context’s physical and psychological aspects. In an online questionnaire study, we measured the following characteristics.

#### 2.3.1. Sociodemographic Characteristics

The respondents were asked to give their age, gender, and education level background information (Table 1). For their place of residence, the respondents indicated the area of their home (in m^2^) and the number of persons living there regularly (Table 2). We did not differentiate between the types of relationships between co-residential persons.

#### 2.3.2. The Place of Security at Home

We assessed the experiences related to the ‘place of security’ with a procedure inspired by the Emotional Map of the Home Interview [30]. The original procedure used an in-person and in-depth interview protocol supported by a drawing process where research participants drew a plan of their homes and identified several emotionally significant places—such as the place of security, insecurity, togetherness, and tension—along with sharing personal stories in the interview about the specific place-related experiences. While several features of this process were impossible to be adapted to an online study, we included the following instructions to use the original procedure’s core features as much as possible.

First, we asked the respondent to recall their home as vividly as possible, leading the imagination with textual prompts through an imaginary entering of their home and going through their spaces. Following this instruction, the respondents could step forward after agreeing with the statement, ‘I could imagine my home; I recalled it’.

Second, we asked the respondents to find a place of security in their homes and to label it with a short description. We intended this instruction to help respondents recall their place of security as vividly as possible. The instruction reads: “Now we will ask you about your experiences in your home. Where is your place of security in your home? Where do you feel most secure in your home? (If you cannot think of any such a place in your home, please describe where you feel most secure outside your home.) Name this place in a few words. To which part of the room/piece of furniture is it related? Briefly describe what happens here when you experience security in this place.”

After finding the place of security, twelve statements were formulated to capture the niche construction processes: shaping through striving for agency and communion and the experienced emotional feedback. All statements were framed with direct reference to the place chosen by the respondents, starting with the stem “When I’m at my secure place….” Response options ranged from 1 (not at all true for me) to 7 (to a great extent true for me). Six statements reflected the agentic and communion aspects of the place experiences at the place of security (e.g., “Security is connected to my activities.” and “Security is connected to my relationship with the people I live with.”, respectively).

The subsequent six statements were adapted from a study on self-regulation in favorite places used by Korpela and colleagues [28], where the items referred to the experiences of visiting one’s favorite place. The items could be divided into two dimensions of experiences: ‘positive recovery of self’ and ‘low self-confidence—distress.’ Based on Korpela and colleagues’ results, we chose six statements with the highest factor loadings from the original item pool that listed cognitive and emotional experiences as self-regulatory processes that the respondent may have had at their favorite place and, in our case, in their place of security. Three statements captured the positive self-recovery experiences in the place and three the experiences related to low confidence and distress (e.g., “I feel like a unique and valuable person at that place.” and “I feel that my self-confidence decreases when I am there.”, respectively; for an overview of the items also see Table 3).

#### 2.3.3. Psychological Characteristics

Participants were instructed to refer to the present (i.e., the pandemic and lockdown-related) experiences when replying about the following characteristics.

##### Basic Psychological Need Satisfaction

Satisfaction of basic psychological needs was measured using the Hungarian version of the Basic Psychological Need Satisfaction Scale [43]. The questionnaire contains nine items that reflect personal experiences with basic psychological need satisfaction and comprise three sub-scales, namely Autonomy (e.g., I generally feel free to express my ideas and opinions), Competence (e.g., Most days I feel a sense of accomplishment from what I do), and Relatedness (e.g., I really like the people I interact with). Items were rated using a 7-point Likert-type scale (1 = Not at all characteristic, 4 = Moderately characteristic, 7 = Fully characteristic). The subscales demonstrated good internal consistency for Autonomy (Cronbach’s α = 0.69; 3 items), Competence (Cronbach’s α = 0.70; 3 items), and Relatedness (Cronbach’s α = 0.75; 3 items).

##### Satisfaction with Life Scale

The Satisfaction With Life Scale (SWLS; Hungarian adaptation [44,45]) is a 5-item measure for assessing overall general satisfaction with life, where respondents indicate their degree of agreement on a 7-point Likert-type response scale (1 = strongly disagree, 7 = strongly agree). A sample item is ‘In most ways, my life is close to my ideal.’ The alpha coefficient indicated good-to-excellent reliability in the sample (α = 0.88).

##### Satisfaction with Home Scale

Using the concept and measurement of life satisfaction, Sallay and Martos developed a scale to measure participants’ satisfaction with their existing home, the Satisfaction With Home Scale [41]. The scale contains the modified items of the original SWLS (e.g., ‘In most ways, my home is close to my ideal’) and uses the same Likert-type response scale (1 = strongly disagree, 7 = strongly agree). The internal consistency estimate was excellent in the sample (α = 0.91).

### 2.4. Analytic Strategy

An exploratory factor analysis was performed on the place-of-security-related characteristics of the agentic, communion, and cognitive-emotional experience items using the SPSS statistical package (Version 24). Principal axis factoring estimation was applied with direct oblimin rotation. To identify the number of factors, we applied parallel analysis. We verified sampling adequacy with Kaiser–Meyer–Olkin measure (KMO > 0.8 as adequate and >0.6 as acceptable) and tested if the correlation matrix was significantly different from an identity matrix by Bartlett’s test of sphericity. We calculated the resulting factor scores based on the Regression method for the next analytic step. Factor scores were all standardized, with a mean of 0.00 and a standard deviation of 1.00. After checking the sample for multivariate outliers, we used the factor scores as indicator variables for Latent Profile Analysis.

Latent Profile Analyses were conducted using the freeware statistical software Jamovi [46] with the module snowRMM [47] that utilizes tidyLPA R-package [48]. The different solutions were compared based on multiple statistical fit indices. Better model fit was indicated by lower levels of the Akaike Information Criterion (AIC), the Bayesian Information Criterion (BIC), and a non-significant bootstrap likelihood ratio test (BLRT) in the successive model. We considered BIC and BLRT the most applicable indices in identifying model fit [49]. We interpreted latent profiles based on their characteristic means on the indicator variables (factor scores). Latent profile membership was used to identify specific subgroups compared in a series of subsequent ANCOVAs controlled for gender, age, and home characteristics.

## 3. Results

### 3.1. Basic Characteristics of the Places of Security

First, we conducted 2 separate factor analyses on 2 randomly assigned subsamples (N = 382 and 375) to test whether the emerging factor solutions were stable. In both analyses, the parallel analysis pointed toward a four-factor solution. Subsequently, factor matrices were rotated toward each other, using Procrustes rotation. Procrustes rotation can be applied to identify factor solutions’ similarity where the structure of the factors has not been tested previously, and the assumptions of confirmatory factor analysis (CFA) are considered too restrictive [50]. The resulting Tucker phi coefficients [51] indicate the factors’ congruence (i.e., their structural similarity) for the two solutions. Phi above 0.90 indicates good, and above 0.95 indicates excellent fit. In our analysis, the Procrustes rotation of the 2 factor matrices yielded phis ranging from 0.90 to 0.98 for the 4 factors. Since parallel analyses and congruence indices of the two random subsamples indicated that the factor structure of the sample was stable, we performed the final exploratory factor analysis on the total sample.

Concurrently, the initial analysis indicated that the 12 items formed 4 reliable factors. The KMO measure (KMO = 0.673) and the result of Bartlett’s test of sphericity (χ2(66) = 1151.5; *p* < 0.001) were acceptable. Similar to the preliminary results in the subsamples, the parallel analysis pointed to 4 nonrandom factors, explaining 54.8% of the variance. After the factor extraction (principal axis factoring) and direct oblimin rotation, the 4 factors explained 34.13% of the total variance (Table 3).

In total, 2 emerging factors (factors 1 and 4) represented a positive recovery of self and distress/low self-confidence in the place, respectively, which corresponded to the emotional experience dimensions found by Korpela and colleagues [28]. The other 2 factors (factors 2 and 3) could be distinguished by the ‘agentic’ and ‘communion’ labels, representing the experiences’ active, agentic, and relational aspects, respectively. However, at the item level, ‘feeling a unique and valuable person’ loaded approximately equal to the positive recovery and the agency factors. Moreover, the item ‘talking to somebody’ loaded on the agency factor and only slightly on the communion factor. We used factor scores in the subsequent latent profile analysis and the resulting comparisons and visualizations. We calculated the factor score reliability indices with the procedure of Beauducel and colleagues [52]. The 4 factors presented acceptable reliability (reliability scores ranging from 0.63 to 0.82). The bivariate correlations between the study variables are presented in Table 2.

### 3.2. Profiles of Secure Place Experiences

The main aim of our study was to determine the place-of-security-related patterns of place experiences. To this end, we used the scores of the four factors (Personal agency, Communion, Positive recovery of self, and Distress) as indicator variables for latent profile analysis. This method can identify subgroups with distinct profiles of place experiences. We applied 4 possible constraints provided by the tidyLPA package on the local distributions (variance) and covariances of the distinct latent profiles: equal variances and 0 covariances (Model 1), varying variances and 0 covariances (Model 2), equal variances and varying covariances (Model 3), and varying variances and varying covariances (Model 6). Latent Profile Analyses with 2 to 8 solutions for the 4 models and fit indices for the consecutive profile numbers are presented in Appendix A. Local minima of BIC were achieved in Model 2 with 4 and 6 latent profiles, confirmed by non-significant BLRT tests in the 5- and 7-profile solutions. A non-significant (*p* > 0.05) result from the BLRT test for a given model with k number of latent profiles indicates that it is more appropriate to retain the model with k-1 latent profiles since the k-profile solution does not provide a significant increase in model fit. Comparing the results, we chose the four-profile solution since this was the first, most parsimonious solution, which also had very similar and low BIC values compared to the six-profile solution. Therefore, we retained the profile membership classifications of four groups for further analysis. Table 4 and Figure 1 present the subgroup profiles using the group means on the initial standardized factor scores.

The groups LP1, LP3, and LP4 are similar in their use of the secure place: their agency and communion are relatively high compared to the lower scores of LP2 on these dimensions. However, the three groups show different profiles for their emotional experiences in the place. Group LP1 is the largest subgroup (N = 213, 28.1%); for them, the secure place has a restorative character (low distress, high self-recovery). This type of secure place experience may be called “security in active self-recovery.” In Group LP3 (N = 207, 27.3% of the sample), agency and communion are connected with medium-level self-recovery and low distress. We may call this type of place experience “security in doing and feeling good enough.” The main characteristic of Group LP4 (N = 145, 19.2% of the sample) is that their emotional experience at their secure place is relatively negative (medium self-recovery coupled with the highest distress). Although they may struggle with their negative feelings, they can also find self-recovery in the place. Therefore, we labeled this place experience “security in stress and compensation.” Finally, respondents in Group LP2 (N = 192, 25.4% of the sample) have the lowest agency and relatively low communion scores. In contrast, their emotional experiences are less favorable: self-recovery is the lowest compared to the other groups, and distress is at a medium level. This group may represent the place experience of “security in detachment” since group members are, to a certain extent, distanced from engaging in activities and restoring their emotions.

Latent profile membership was cross-tabulated with gender, education, and living alone (vs. with others) in the home. Education was not related to group membership (Chi-square = 5.71 (df = 6), *p* = 0.456), nor were participants living alone more likely members of any of the groups (Chi-square = 4.31 (df = 6), *p* = 0.230). Concerning gender, the overall Chi-square test indicated a significant difference in the distribution of membership ratios (Chi-square = 12.32 (df = 3), *p* = 0.006). Detailed inspection indicated that male respondents were likelier to be members of Group LP2 while less likely to be members of Group LP1.

Latent profile groups were also compared across age, the home floor area, and the number of people living there. No significant difference could be detected between the groups concerning the floor area (F (3) = 1.05, *p* = 0.370) and the total number of inhabitants (F (3) = 0.45, *p* = 0.720). Concerning age, the overall F test indicated a significant effect (F (3) = 6.74, *p* < 0.001) with Groups LP1 and LP3 being older (M = 39.49, SD = 13.21, and M=40.11, SD = 12.99, respectively). Post hoc tests with Bonferroni correction indicated that Groups LP2 and LP4 were significantly younger (M = 35.74, SD = 9.95 and M = 36.32, SD = 12.05, respectively).

### 3.3. Predictors of the Profile Group Membership

We assumed that better place-related self-regulation experiences would associate with better psychological functioning in the broader environment. We used multinomial logistic regression to predict latent profile groups by gender and age (as these were significant correlates of the profiles in bivariate analysis) and a series of psychological characteristics that captured information about the person’s perception of the broader niche. The psychological characteristics included basic psychological need satisfaction during the lockdown, such as the extent of autonomy, competence, and relatedness experiences, and satisfaction with one’s life in general and with one’s home in particular. We chose Group LP3 (“security in doing and feeling good enough”) as a reference category since we considered this group the average type. Detailed results are presented in Table 5.

Membership in Group LP1 (“security in active self-recovery”) was predicted by increased felt competence (OR = 1.22, *p* = 0.046) compared to Group LP3. Members in Group LP2 (“security in detachment”) were more likely to be younger and male respondents (OR = 0.97, *p* < 0.001, and OR = 1.67, *p* = 0.031, respectively) with lower felt competence and relatedness (OR = 0.79, *p* < 0.016, and OR = 0.75, *p* = 0.039, respectively). Finally, compared to Group LP3, members of Group LP4 were younger, experienced less relatedness in their life, and were less satisfied with their life (OR = 0.70, *p* < 0.018, and OR = 0.70, *p* = 0.010, respectively). Felt autonomy and satisfaction with home did not predict any latent profile memberships in the analysis.

## 4. Discussion

In the present study, we explored meaningful patterns of home-related experiences in places of security for Hungarian respondents during the COVID-19 lockdown. We also addressed the associations of well-being characteristics with the experiences in their secure places. Our results capture a specific transitory period in our respondents’ lives in their family homes. On the one hand, they refer to a specific period in the COVID-19 pandemic, focusing on the dynamically changing first lockdown. On the other hand, they may also represent different stages in the self-regulation process, which may have different timing for different people. With all these aspects in mind, our results provided valuable insight into the niche construction processes of striving for security during the first phase of the COVID-19 pandemic.

### 4.1. Dimensions of Secure Place Experiences

The personal characteristics of place use and, on the other hand, emotional experiences in the place of security could be distinguished in the factor analysis of the items representing secure place experiences. Two distinct factors (agency and communion) describe the behavioral aspects of place use. In comparison, two factors (positive recovery of the self and distress) represent the emotional valence of the place of security. Importantly, we found low intercorrelations between the four factors, indicating that these characteristics are conceptually different and cannot be deducted from each other. The analysis replicated two essential dimensions of the experience in places of security [28]. Similarly, other studies indicated that place-related activities might have a double character involving self-identity and fun along with experiences of stress even in personally important places [23]. In some instances, the distress associated with places of security was a salient feature of the responses we will discuss later.

### 4.2. Types of Secure Place Experiences and Well-Being

Using latent profile analysis, we identified four types of experiences in the place of security. Moreover, we analyzed the possible associations with how members of a particular subgroup may have specific perceptions about the broader context: characteristics of the person’s social ecology, the level of basic psychological need satisfaction (autonomy, competence, and relatedness) [5,6], and the satisfaction with environmental resources (life in general and home) [40,41]. We will interpret the emergent latent profile patterns together with their context.

Three groups, LP1, LP3, and LP4, showed similar place-related agency and communion profiles. Compared to Group LP2, they were relatively active and experienced slightly more communion in their place of security. However, their emotional experiences differed considerably. We interpreted Group LP3 as an average functioning, “doing and feeling good enough” subsample, with average self-recovery and low distress. Using Group LP3 as a reference group, we found that Group LP1 (“security in active self-recovery”) had the highest self-recovery. Moreover, their signature characteristic was a higher satisfaction concerning the need for competence. These associations strengthen the interpretation that respondents in Group LP1 were capable of active, skillful, and resilient coping with the adversities of the lockdown. They could capitalize on the use of their places of security for self-restoration. Interestingly, no other characteristics predicted the membership in this group, suggesting that this outcome is primarily the function of self-regulation competence.

On the contrary, members of Group LP4 experienced security in “stress and compensation.” This group, including every fifth respondent in the sample, evidences a certain level of struggle for functional self-regulation that may also partly overwhelm their experiences in the place of security. General stress and challenges might also be present in the lives of these respondents, which can be inspected in lower relatedness and life satisfaction being predictors of membership in this group. According to the results, these respondents might use their places of security to let their distress out, but without an effective instant downregulation.

The members of Group LP2 (“security in detachment”) engage less in activities and relationships. Although place-related distress is low in this group, they do not fully capitalize on the positive self-regulatory potential of their transactions with their safe place. This association may be partly explained by their lower competence and relatedness experiences. Less satisfied needs for competence and relatedness may cause them to disengage from activities and relationships in their places of security.

Overall, these results confirm that place of security experiences have complex patterns that cannot be deducted from the linear associations among the individual dimensions. For example, while higher agency and communion correlated positively with life satisfaction, Group LP4 was as high in agency and communion as other groups, and group membership was predicted by lower life satisfaction. The possibility of detecting complex, nonlinear associations at a systemic level is one of the strengths of the person-oriented research approach [38].

Moreover, the results also show that perceptions of the actual life conditions (i.e., need satisfaction and satisfaction with life) are superior predictors of place-based self-regulation processes compared to objective characteristics such as the area of the flat or the number of persons living there. Although previous research found that adverse physical conditions might have been detrimental to mental health during the pandemic [8], our results show that the subjective perception of these contextual factors may partly transmit this effect.

It is important to note that the groups LP2 and LP4, including almost half of the sample, present two specific vulnerable types of place experience. They may lack adequate resources in life and support for their basic psychological needs. Although we have no reference estimates from before the pandemic, the high proportion of respondents in these two groups may signify the societal-level crisis caused by the COVID-19 pandemic. Since these groups were associated with younger ages (LP2 and LP4) and partly being male (LP4), the results also highlight that broad social groups may need attention in these adverse situations. Youth might be at particular risk [13,53].

### 4.3. Distress in Secure Places: A Sign of Vulnerability or Slow Recovery?

Previous research demonstrated that distress might be present even in specific places of security, such as home environments [28]. In their interpretation, negative experiences in valued places indicate a failure of self-regulation or a cross-sectional snapshot of a longer recovery process. Our sample’s “stress and compensation” group demonstrates this possibility. However, top-down factors may also affect membership in the vulnerable subgroup, such as mental states associated with subclinical depression or negative cognitive styles [54].

On the other hand, a place of security may provide an opportunity for a slow recovery for people vulnerable to negative mental states. Previous studies showed that individuals who are more anxious in social situations prefer an environment where they do not have to engage in social interactions and where the situation is familiar and safe [55]. The place’s attractiveness is related to the fact that it provides the person with a safe and predictable social environment [56]. Recently, Sallay and colleagues [57] have also found that higher social anxiety predicted being more likely to visit a person’s favorite place for self-restoration. This association emerged for positive and negative reasons for visiting the place (i.e., because of experiencing joy or sadness).

Although vulnerable individuals may experience negative feelings there, the place of security may serve as a way to achieve positive self-regulation. We may assume that it has a similar effect as the “safe/comfortable/quiet place” technique in EMDR therapy [58] and psychodrama [59]. These techniques serve as a tool for anxiety relief and trauma processing. Similarly, nurturing the internal image of a ‘safe place’ and finding a ‘place of security’ in one’s home can be mutually reinforcing experiences, which, in turn, may lead to a self-restoring effect of the home environment for distressed individuals.

### 4.4. Place of Security at Home as Niche Construction

Our study also confirmed the notion of the psychological home concept [60]: home experiences have cognitive, affective, and behavioral references to security and belonging. Places of security represent a dynamic, temporally, and spatially constructed space, a “personal niche” for activating adaptive modes of emotion regulation and thus maintaining mental equilibrium [20,23,61,62]. Places of security may evoke intense positive or negative emotional experiences [30]. The prevalence of positive/restorative or negative/distressful experiences depends on the effectiveness of environmental self-regulation processes. The types of secure place experiences may reflect the individual characteristics of the person (their mental state and the level of environmental competence), the ongoing relationship processes, and the level of fit between the person’s preferred activities and the physical characteristics of the place. Thus, niche construction in the space of the home involves strategies of self-recovery through the experience of agency for specific individuals. In contrast, for others it involves nurturing relationships and, for others again, creating a space for being alone. Some niche construction strategies are connected to more vulnerable states of well-being, such as disengaging from others in one’s place of security or struggling to compensate for a higher stress level.

### 4.5. Limitations

Our research has several limitations that need to be regarded when interpreting our results. First, cross-sectional data collection does not allow us to infer causal relationships. Moreover, cross-sectional design can show only a snapshot, while self-regulation is a dynamic process. In the future, it may be worthwhile to focus on the temporal dynamics of using specific places at home and the association with self-regulation outcomes.

Second, online survey data collection includes the shortcomings of self-report methods. Moreover, some potential confounders, such as mental health status, were excluded only by self-report. Although we used general life satisfaction, a well-established measure of positive functioning, later studies should use a refined assessment of psychopathology.

Third, we relied only on Likert-scale ratings in our analysis. At the same time, the Emotional Map of the Home Interview procedure may be used as an interview [30], providing rich and meaningful data for interpretation. Advanced use of our procedure may include the analysis of the narratives concerning the secure place experiences.

Finally, we combined several newly developed item ratings about places of security in the analysis. The consistent, well-interpreted pattern of responses suggests that the dimensions of experiences associated with these places can be generalized. However, subsequent studies are needed to confirm the measurement of self-regulation processes and outcomes in places of security at home.

## 5. Conclusions

Our study aimed to identify types of security experiences at home during the first COVID-19 lockdown in a sample of Hungarian respondents. The main finding of our study is that we could present the multiple faces of striving for and finding psychological security at home. Our study was the first to explore home experiences during the COVID-19 lockdown in a pattern-oriented approach. These associations highlighted the primary role of psychological processes that shape home experiences and the notion of psychological home. Moreover, our results may help mental health professionals and practitioners identify potentially vulnerable subgroups. At the same time, we recommend including practices of place-based self-regulation in the focus of professional exploration and interventions. Home as a dynamic, temporally, and spatially constructed space, a “personal niche,” may support psychological stability during transitions and novel and adverse life circumstances.

## Figures and Tables

**Figure 1 behavsci-13-00009-f001:**
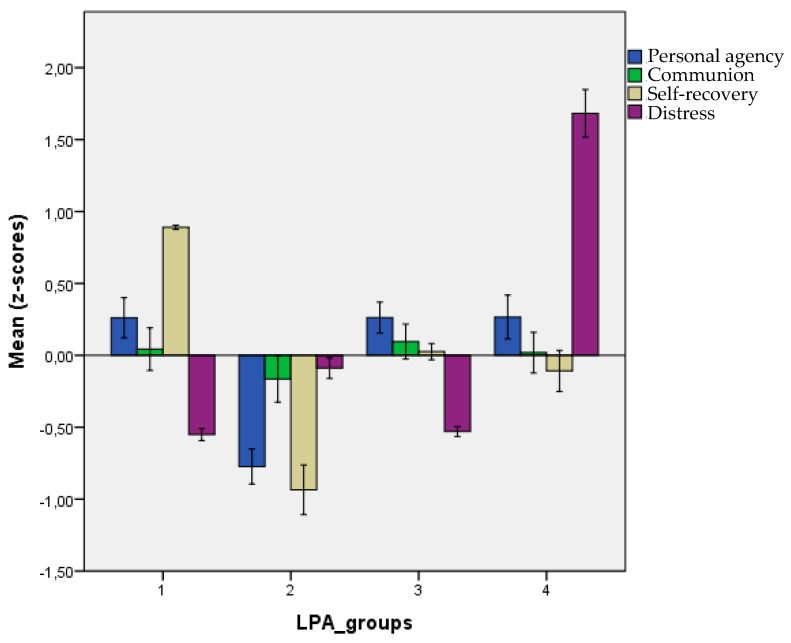
Profiles of the latent profile groups. Note: LP1 = “security in active self-recovery”; LP2 = “security in retreat”; LP3 =“security in doing and feeling good enough”; LP4 = “security in stress and compensation”. Dimension scores are standardized factor scores with +/−2 SE error bars.

**Table 1 behavsci-13-00009-t001:** Sociodemographic characteristic of the participants.

	N	%
Gender		
Male	218	28.8
Female	539	71.2
Education		
primary	68	9.0
secondary	162	21.4
BA or higher	527	69.6
Living alone vs. with others		
alone	33	4.4
with others	724	95.6
N total	759	100

**Table 2 behavsci-13-00009-t002:** Bivariate associations between study variables.

				Correlations										
		M	SD	1	2	3	4	5	6	7	8	9	10	11
1	Personal agency	0.00	1.00											
2	Communion	0.00	1.00	0.11 **										
3	Positive recovery of self	0.00	1.00	0.16 ***	0.00									
4	Distress	0.00	1.00	0.02	−0.03	−0.13 ***								
5	Age (years)	38.03	12.31	0.24 ***	0.02	0.05	−0.13 ***							
6	N (home)	2.97	1.28	0.04	0.10 **	−0.03	−0.01	0.06						
7	Floor area	92.94	51.22	0.08 *	0.01	0.05	−0.05	0.16 ***	0.45 ***					
8	Autonomy	4.92	1.26	0.16 ***	0.00	0.12 **	−0.22 ***	−0.06	−0.04	0.02				
9	Competence	4.68	1.33	0.24 ***	−0.02	0.16 ***	−0.17 ***	0.04	0.03	0.08 *	0.52 ***			
10	Relatedness	5.94	0.97	0.19 ***	0.07	0.11 **	−0.20 ***	−0.07	−0.03	0.02	0.51 ***	0.32 ***		
11	SWLS	5.12	1.12	0.15 ***	0.16 ***	0.08	−0.25 ***	−0.01	0.06	0.15 ***	0.41 ***	0.39 ***	0.36 ***	
12	SWHS	4.95	1.34	0.19 ***	0.11 **	0.12 **	−0.15 ***	0.15 ***	0.07	0.30 ***	0.31 ***	0.29 ***	0.29 ***	0.53 ***

Note: personal agency, communion, positive recovery of self, and distress variables are factor scores drawn from principal axis factoring with direct oblimin rotation. N (home) = number of people living in the home; SWLS = Satisfaction with Life Scale; SWHS = Satisfaction with Home Scale. * *p* < 0.05; ** *p* < 0.01; *** *p* < 0.001.

**Table 3 behavsci-13-00009-t003:** Principal axis factoring of the place of security attributes.

	Factors			
Item	1	2	3	4
I can dream and wish to accomplish personally important and pleasant aspirations.	**0.81**	0.10	0.02	−0.11
I can ponder future threats or problems and anticipate solutions to them.	**0.66**	0.17	−0.05	−0.06
I am active.	0.05	**0.53**	0.05	−0.04
I am talking to others.	0.07	**0.50**	−0.09	0.12
I feel I am a unique and valuable person.	**0.41**	**0.43**	0.17	−0.16
Security is connected to my activities.	0.15	**0.43**	0.15	0.07
I have an impact on what happens there.	0.18	0.29	0.07	−0.28
I am with others (vs alone).	−0.03	−0.08	**0.68**	−0.02
Security is connected to my relationship with the people I live with.	0.06	0.30	**0.64**	0.01
I feel that my self-confidence decreases.	−0.04	0.04	−0.02	**0.48**
Being there feels distressing.	−0.07	0.08	−0.02	**0.47**
I feel a failure there.	−0.01	−0.03	0.02	**0.32**
Explained variance %	*16.02*	*7.98*	*6.14*	*4.00*
Factor score reliability	*0.82*	*0.63*	*0.64*	*0.63*

Note: factor loadings above |0.30| are presented in bold.

**Table 4 behavsci-13-00009-t004:** Comparison of the latent profile groups along the place characteristics.

		Personal Agency		Communion		Positive Recovery		Distress	
	N (%)	m (SD)	95% CI	m (SD)	95% CI	m (SD)	95% CI	m (SD)	95% CI
LP1	213 (28.1%)	0.26 (1.02)	0.12, 0.40	0.04 (1.08)	−0.11, 0.19	0.89 (0.11)	0.88, 0.90	−0.55 (0.30)	−0.59, −0.51
LP2	192 (25.4%)	−0.77 (0.84)	−0.89, −0.65	−0.16 (1.12)	−0.32, 0.00	−0.93 (1.20)	−1.10, −0.76	−0.09 (0.49)	−0.16, −0.02
LP3	207 (27.3%)	0.26 (0.78)	0.15, 0.37	0.10 (0.88)	−0.02, 0.22	0.03 (0.41)	−0.03, 0.09	−0.53 (0.24)	−0.56, −0.50
LP4	145 (19.2%)	0.27 (0.91)	0.12, 0.42	0.02 (0.85)	−0.12, 0.16	−0.11 (0.86)	−0.25, 0.03	1.68 (1.00)	1.52, 1.84
F (overall)		64.20		2.50		203.40		606.99	
*p*		<0.001		0.056		<0.001		<0.001	
eta^2^		0.204		0.010		0.448		0.707	
post hoc ^a^		LP1, LP3, LP4 > LP2		LP1, LP3, LP4 > LP2		LP1 > LP3, LP4 > LP2		LP4 > LP2 > LP1, LP3	

Note: ^a^ post hoc test with Bonferroni adjustment. LP1 = “security in active self-recovery”; LP2 = “security in detachment”; LP3 = “security in doing and feeling good enough”; LP4 = “security in stress and compensation”.

**Table 5 behavsci-13-00009-t005:** Multiple logistic regression of the latent profile group membership.

	Latent Profile Membership (Reference Group: LP3, N = 182)											
	LP1				LP2				LP4			
	OR	95% CI Lower	Upper	*p*	OR	95% CI Lower	Upper	*p*	OR	95% CI Lower	Upper	*p*
Male (vs. Female)	0.66	0.41	1.07	0.090	**1.67**	**1.05**	**2.65**	**0.031**	1.22	0.73	2.03	0.457
Age (years)	0.99	0.98	1.01	0.537	**0.97**	**0.95**	**0.98**	**<0.001**	**0.97**	**0.95**	**0.99**	**0.004**
Autonomy	0.99	0.79	1.24	0.909	0.86	0.69	1.08	0.189	0.83	0.64	1.06	0.129
Competence	**1.22**	**1.00**	**1.48**	**0.046**	**0.79**	**0.65**	**0.96**	**0.016**	1.05	0.85	1.31	0.638
Relatedness	0.87	0.66	1.16	0.347	**0.75**	**0.57**	**0.99**	**0.039**	**0.70**	**0.52**	**0.94**	**0.018**
Life satisfaction	1.04	0.81	1.34	0.744	1.26	0.98	1.62	0.071	**0.70**	**0.54**	**0.92**	**0.010**
Home satisfaction	1.06	0.86	1.29	0.598	0.84	0.69	1.02	0.077	1.09	0.88	1.36	0.437
N	186				178				123			

Note: LP1 = “security in active self-recovery”; LP2 = “security in detachment”; LP3 = “security in doing and feeling good enough”; LP4 = “security in stress and compensation”. Total N = 669 for this analysis due to missing data in the predictors. Significant (*p* < 0.05) predictors are in bold.

## Data Availability

The data presented in this study are available on request from the corresponding author. The data are not publicly available due to privacy concerns.

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
