# Peer review of "Finding a Secure Place in the Home during the First COVID-19 Lockdown: A Pattern-Oriented Analysis"

_behavsci, 2022, doi:10.3390/bs13010009_

Round 1

Reviewer 1 Report

The  manuscript deals with the analysis of possible profiles of secure experiences and feeling during pandemic lockdown, in a large community sample. I think its content can fit Behavioral Sciences aims.

As critical aspects, just a few methodological questions and a concern about the extent of results.

-          First of all, I think the first part of the title (“Secure place experiences in the home…”) can lead to find an erroneus  content, because data are related with the search of secure experience profiles and its relationship with determinated psychological variables, and not with specific secure experiences.

-          Sample was obtained via an online procedure. No more information is provided (and there is not a ‘procedure’ subsection). Can authors provide a more detailed information?

-          To measure “secure places experiencies” author(s) developed an inventory. This inventory yielded four factors. Despite a parallel analysis as was performed, I suggest o run at least two factor analysis (with two random subsamples), to determine the stability of the type and number of factors. Also, please, provide internal consistencies of those factors. How author(s) explain common variance only reached a 34%?

-          I feel confused about the decision of a five latent profile solution. Why that solution is better than the three profile solution? According which parameters?

-          And my major concern is about the scope of data obtained: the secure experience profiles and their relationship with psychological variables have something like obvious. I miss a deeper discusion about the implications of these results, including social a conceptual implications (as it is point out in 4.4. discusion subsection). What this document adds to the knowledge in the field?

Author Response

We are thankful for the Reviewer’s critical comments, which made us considerably rewrite the methodological part of the study and extend the scope of our reasoning. Below we present how we dealt with the comments. Also note that we thoroughly revised the manuscript for inconsistent word use and corrected the text accordingly.

The  manuscript deals with the analysis of possible profiles of secure experiences and feeling during pandemic lockdown, in a large community sample. I think its content can fit Behavioral Sciences aims.

As critical aspects, just a few methodological questions and a concern about the extent of results.

-          First of all, I think the first part of the title (“Secure place experiences in the home…”) can lead to find an erroneus  content, because data are related with the search of secure experience profiles and its relationship with determinated psychological variables, and not with specific secure experiences.

We reconsidered the title, which reads now as “Finding a secure place in the home during the first COVID-19 lockdown: a pattern-oriented analysis”

-          Sample was obtained via an online procedure. No more information is provided (and there is not a ‘procedure’ subsection). Can authors provide a more detailed information?

We extended the Procedure subsection with more information on the recruitment process.

-          To measure “secure places experiences” author(s) developed an inventory. This inventory yielded four factors. Despite a parallel analysis as was performed, I suggest to run at least two factor analysis (with two random subsamples), to determine the stability of the type and number of factors. Also, please, provide internal consistencies of those factors. How author(s) explain common variance only reached a 34%?

Thank you for this suggestion: our measure was developed for this study, and we agree that multiple validations of its properties are essential.

We ran two FA-s on two randomly selected subsamples. Both subsamples provided four factors. Moreover, the factor matrices could be rotated toward each other with high congruency via Procrustes rotation.

We also provided factor score reliability estimates, which were acceptable.

Finally, we provided more information on factor extraction. The first four factors explained 55% of the total variance, which we deem acceptable. However, we used principal axis factoring to determine the final factor solution and the factor scores. PAF procedure estimates the error in the data, and the resulting explained variance refers only to the proportion of “true” variance that is likely free of error.

We included the above information in the text.

-          I feel confused about the decision of a five latent profile solution. Why that solution is better than the three profile solution? According which parameters?

Decisions on the number of latent profiles may follow objective (i.e., indices) and subjective aspects (i.e., interpretability, meaningfulness). We considered that three and five-profile solutions provided almost identical fit indices, while the five profiles were more complex and still comprehensive.

However, we wanted to take the comment seriously and investigate the solutions more deeply. Therefore, we reran the analyses in a newly published, extended version of the Jamovi statistical package where multiple configurations of constraints can be defined (i.e., whether the subgroups’ variances are equal or varying and covariances are zero, equal or varying). This analysis showed that the earlier model was somewhat misspecified, and a better fitting solution can be found using Model 2 (i.e., variances are varying and covariances are zero). Again, there were two solutions with very similar goodness of fit indices. We chose the four-profile solution for further analysis since it was the more parsimonious solution with adequate complexity.  

-          And my major concern is about the scope of data obtained: the secure experience profiles and their relationship with psychological variables have something like obvious. I miss a deeper discusion about the implications of these results, including social a conceptual implications (as it is point out in 4.4. discusion subsection). What this document adds to the knowledge in the field?

We considered this comment critical to our research approach and provided the following reasoning and justification in the revised manuscript.

We interpreted psychological characteristics as part of the context people strive to find a secure place at home. The satisfaction of basic psychological needs has been long considered the indicator of the social environment’s supportiveness in Self-determination Theory. Moreover, satisfaction with one’s life in general, and home in particular represent the person’s subjective (cognitive) evaluation of the personally important resources for a good life. In the revised manuscript, we assumed that these psychological characteristics represent the person's current evaluations of the lockdown situation. These evaluations may, in turn, affect the striving for finding security, the associated self-regulatory efforts, and their actual patterns at home.

Accordingly, we reconsidered the necessary analyses and treated general psychological well-being measures as predictors of latent profile membership. Obviously, our cross-sectional design doesn’t allow causal inferences and in complex social-ecological and psychological systems like homes, contextual factors (evaluation of available psychological resources in the broader environment) may also be affected by lower-level processes (self-regulation in the secure place).

Reviewer 2 Report

The authors present an important topic "Secure place experiences in the home during the first COVID-19 lockdown: a pattern oriented analysis"

However, the paper has a few shortcomings.

1. The abstract is insufficient. It must be improved, add the aim of the study and major results.

2. The aim of the study could be highlighted in the method section too. The same goes for the hypothesis of the study.

3. What is the impact of the study limitation on your study and how did you mitigate it?

The study conclusion must be redone. It is nothing close to a study conclusion. See how previous studies concluded. Take the citations there to the discussion section and write new conclusions, including the aim of the study, major study outputs and recommendations.

Author Response

We are thankful for the comments and suggestions of the Reviewer. Below we provide our answers to the raised points. Please take into account that after considering the comments of Reviewer 1, we considerably restructured our reasoning and analyses.

Specifically, we reanalyzed the data with an improved statistical procedure and as a result, we accepted and interpreted a four-profile latent profile solution. We found evidence that this solution represents better and more parsimoniously the latent structure of the data. Moreover, we treated indices of psychological well-being as contextual factors for place-based self-regulation efforts and not as their outcomes. We present the revised reasoning and the corresponding analytical approach in the revised manuscript. Also, note that we thoroughly revised the manuscript for inconsistent word use and corrected the text accordingly.

The authors present an important topic "Secure place experiences in the home during the first COVID-19 lockdown: a pattern oriented analysis"

However, the paper has a few shortcomings.

  1. The abstract is insufficient. It must be improved, add the aim of the study and major results.

We completed the abstract with more information on the required topics. We also modified it according to the new results.

  1. The aim of the study could be highlighted in the method section too. The same goes for the hypothesis of the study.

We incorporated the aims and hypotheses in the descriptions of the Method and Results section.

  1. What is the impact of the study limitation on your study and how did you mitigate it?

We included more reasoning on the role of the study’s limitations and the potential ways of mitigating them in future research in the Limitations section.

The study conclusion must be redone. It is nothing close to a study conclusion. See how previous studies concluded. Take the citations there to the discussion section and write new conclusions, including the aim of the study, major study outputs and recommendations.

We reformulated the Discussion section and, specifically, the conclusions by adding more reflection on the study’s main points and relevance.

Reviewer 3 Report

Overall, the paper is well written, structured and clear. 

The topic is interesting and current, and mostly it can be an innovative contribution on the issue of housing.  The abstract is clear and contains the salient elements of the article. 

The procedure is clearly argued as the results, which are summarized in tables that are significant for the readability of the work. In the discussion, the results are interpreted on the basis of the reference literature in a comprehensive way. Possible future perspectives are also reported in the conclusions.

While I have no major comments, I only have a suggestion on the possibility of including in the introduction some recent and valuable contributions on the subject matter.

See for example:

- Romoli, V., Cardinali, P., Ferrari, J. R., & Migliorini, L. (2022). Migrant perceptions of psychological home: A scoping review. International Journal of Intercultural Relations, 86, 14-25. https://doi.org/10.1016/j.ijintrel.2021.10.009

- Amerio, A., Brambilla, A., Morganti, A., Aguglia, A., Bianchi, D., Santi, F., & Capolongo, S. (2020). COVID-19 lockdown: housing built environment’s effects on mental health. International journal of environmental research and public health, 17(16), 5973.https://doi.org/10.3390/ijerph17165973

 Moreover, a second observation concerns the sample, in fact I suggest that you should include the measures related to the sample characteristics in the results section.

Author Response

We are thankful for the comments and suggestions of the Reviewer. Below we provide our answers to the raised points. Please take into account that after considering the comments of Reviewer 1, we considerably restructured our reasoning and analyses.

Specifically, we reanalyzed the data with an improved statistical procedure and as a result, we accepted and interpreted a four-profile latent profile solution. We found evidence that this solution represents better and more parsimoniously the latent structure of the data. Moreover, we treated indices of psychological well-being as contextual factors for place-based self-regulation efforts and not as their outcomes. We present the revised reasoning and the corresponding analytical approach in the revised manuscript. Also, note that we thoroughly revised the manuscript for inconsistent word use and corrected the text accordingly.

Overall, the paper is well written, structured and clear.

The topic is interesting and current, and mostly it can be an innovative contribution on the issue of housing.  The abstract is clear and contains the salient elements of the article.

The procedure is clearly argued as the results, which are summarized in tables that are significant for the readability of the work. In the discussion, the results are interpreted on the basis of the reference literature in a comprehensive way. Possible future perspectives are also reported in the conclusions.

While I have no major comments, I only have a suggestion on the possibility of including in the introduction some recent and valuable contributions on the subject matter.

See for example:

- Romoli, V., Cardinali, P., Ferrari, J. R., & Migliorini, L. (2022). Migrant perceptions of psychological home: A scoping review. International Journal of Intercultural Relations, 86, 14-25. https://doi.org/10.1016/j.ijintrel.2021.10.009

- Amerio, A., Brambilla, A., Morganti, A., Aguglia, A., Bianchi, D., Santi, F., & Capolongo, S. (2020). COVID-19 lockdown: housing built environment’s effects on mental health. International journal of environmental research and public health, 17(16), 5973.https://doi.org/10.3390/ijerph17165973

We are thankful for the suggestions: we included the indicated papers in the reasoning.

 Moreover, a second observation concerns the sample, in fact I suggest that you should include the measures related to the sample characteristics in the results section.

Thank you for this suggestion. After consulting previous articles in the Special Issue, we considered the presentation of detailed descriptive statistics of the sample more appropriate in the Method section. However, we restructured the presentation of the sample characteristics by adding a new Table (Table 1) where we give details about this data. This way, the reader can get a quick overview of the sample characteristics.

For the same reason, we placed Table 2. (Descriptive statistics and bivariate correlations) in the Method section.

Round 2

Reviewer 1 Report

I think authors have feasibility completed all my major concerns. Consequently, I think manuscript can be accepted.

Author Response

Thank you very much for your feedback and helpful comments.

Reviewer 2 Report

The authors improved the paper accordingly. However, in your final steps, improve your conclusions by stating the aim of the study. As the current conclusion focus on the results and way forward.

Author Response

Thank you very much for your feedback and helpful comments. 

We included a sentence about the aim of the study in the Conclusion section. We also let proofread the manuscript and made minor word use, spelling, and punctuation modifications.